# Vitamin D Deficiency as a Risk Factor for Myocardial Ischemia

**DOI:** 10.3390/medicina57080774

**Published:** 2021-07-29

**Authors:** Christina Batsi, Evangelia Gkika, Loukas Astrakas, Athanassios Papadopoulos, Ioannis Iakovou, Alexandros Dogoritis, Andreas Fotopoulos, Chrissa Sioka

**Affiliations:** 1Department of Nuclear Medicine, University Hospital of Ioannina, 45500 Ioannina, Greece; tinchen58@hotmail.com (C.B.); evagkika@yahoo.gr (E.G.); professor.fotopoulos@yahoo.com (A.F.); 2Department of Medical Physics, University Hospital of Ioannina, 45500 Ioannina, Greece; astrakas@uoi.gr (L.A.); thanospapadopoulos400@gmail.com (A.P.); 32nd Nuclear Medicine Laboratory, AHEPA University Hospital, Aristotle University of Thessaloniki, 54636 Thessaloniki, Greece; iiakovou@icloud.com; 4Neurosurgical Institute of Ioannina, University of Ioannina, 45500 Ioannina, Greece; adogoritis@yahoo.com

**Keywords:** vitamin D, myocardial perfusion imaging, myocardial ischemia

## Abstract

*Background and Objectives:* Vitamin D (Vit D) deficiency has been implicated in various conditions, including cardiovascular disease. The purpose of this retrospective study was to investigate the incidence of patients with myocardial ischemia in relation to their serum levels of vitamin D. *Materials and Methods*: A 64-month search (January 2016 to April 2021) in our database of the Nuclear Medicine Laboratory revealed 113 patients who had both myocardial perfusion imaging with single photon emission computed tomography (MPI SPECT) and Vit D measurements. MPI SPECT obtained myocardial images during both stress (summed stress score, SSS) and rest (summed rest score, SRS). Abnormal MPI SPECT was when the SSS was ≥4. Vit D was determined by radioimmunoassay (RIA). Patients with Vit D values <10 ng/mL, 10–29 ng/mL and ≥30 ng/mL were defined as having a deficiency, insufficiency and sufficiency, respectively. *Results:* Among patients, 46/113 (40.7%) were male and 67/113 (59.3%) were female. Abnormal MPI was found in 58/113 (51.3%) patients. Vit D deficiency was noted in 20/113 (17.7%) patients, insufficiency in 86/113 (76.1%) patients, and normal Vit D was noted in only 7/113 (6.2%) patients. Sixteen of the 20 patients (80%) with Vit D deficiency, and 38/86 (44.2%) with insufficiency had an abnormal MPI SPECT. In contrast, only 1/7 (14.3%) patients with sufficient Vit D levels had an abnormal MPI SPECT. The Mann-Whitney U-test showed that ischemia reduced the values of Vit D. Correlation analysis showed a negative association of Vit D levels with SSS (rho = −0.232, *p* = 0.014) and SRS (rho = −0.250, *p* = 0.008). Further evaluation with a Vit D cut off 20 ng/mL retrieved no statistical significance. Finally, Vit D and gender were independently associated with myocardial ischemia. *Conclusions*: Low Vit D levels may represent a risk factor for myocardial ischemia.

## 1. Introduction

A variety of cardiac biomarkers exist that may be used to diagnose acute coronary syndromes. These biomarkers include creatine kinase myocardial-specific isoenzyme (CK-MB), myoglobin, troponin I and brain natriuretic peptide (BNP) [1]. Utilizing these biomarkers, acute myocardial infarction may be diagnosed safely only 6 h after the event [2]. In addition, serum troponin levels may be useful to assess the risk of sudden cardiac death. [3].

Vitamin D (Vit D) deficiency is present in up to 50% of the general population [4]. Several studies have investigated the implications of low Vit D in several conditions, such as multiple sclerosis (MS), where Vit D consist of a marker for bone loss [5,6]. Low bone mineral density (BMD) may be affected by several parameters in healthy individuals, as well as in patients with various diseases [7,8,9,10,11,12,13,14]. Furthermore, the role of low BMD in patients with coronary artery disease has been well studied, where a correlation was described between low BMD and the number of constricted coronary arteries, indicating that low BMD may represent an independent risk factor for CAD [15,16,17,18]. Thus, low BMD and CAD seem to correlate and share a common implicating marker, and that’s Vit D [5,19].

The etiology of these associations between Vit D and various conditions stems from the fact that Vit D has an impact on multiple biologic pathways important to autoimmunity, cancer, cardiovascular disease, infectious disease, and other conditions [19].

Low Vit D levels may result in vascular inflammation, coronary arterial dysfunction and postinfarction complications [20], as well as adverse cardiovascular outcomes [4]. Apart from the traditional cardiovascular risk factors, Vit D deficiency may predict worse coronary artery disease, and even an increased number of affected coronary arteries [21]. Vit D deficiency may even be the etiology of myocardial dysfunction in patients with silent ischemia and no known history of coronary artery disease [22].

According to the European Society of Cardiology (ESC) new published guidelines in symptomatic patients, a non-invasive procedure is considered a proper initial investigation [23]. Myocardial perfusion imaging (MPI) with single photon emission computed tomography (SPECT), is a noninvasive functional imaging method used in several diseases for myocardial evaluation [24,25,26]. MPI can image myocardial ischemia in patients even in the absence of specific cardiac symptomatology [27,28,29]. The diagnostic accuracy of MPI SPECT compared with conventional coronary angiography is similar for both men and women, with a mean sensitivity and specificity of 84.2% and 78.7% for SPECT MPI in women, and 89.1% and 71.2% for MPI SPECT in men, respectively [30]. 

The aim of the present study was to evaluate whether the variations in the Vit D levels are linked to myocardial ischemia.

## 2. Materials and Methods

### 2.1. Participants and Procedures

We retrospectively reviewed in the medical records of the Nuclear Medicine Department of our University Hospital, all the MPIs of patients between January 2016 and April 2021 (a total of 64 months), which also had a Vit D measurement within 6 months prior or post MPI. No exclusion criteria were used, concerning the etiology of the MPI exam, their medical history, or their cardiovascular status. Our search retrieved 113 MPIs of 67 female and 46 male Caucasian patients that also had Vit D measurement. Their mean age was 64.9 years, 63.7 years for males and 65.7 years for females. 

The patients underwent stress/rest MPI SPECT according to previously published guidelines [31]. MPI evaluation and interpretation was performed independently by two experienced nuclear medicine physicians, blinded to the final diagnosis, and possible disagreements were solved by consensus. SPECT findings were based on a 17-segment model and scored separately for the stress (SSS) and rest (SRS) images, as described previously [32]. A summed difference score (SDS), calculated by subtracting the SRS from the SSS, consisted of a semiquantitative measure of the severity of the reversible ischemic myocardial perfusion defects. Aberrant MPI SPECT denoting ischemia was deemed to be when SSS was ≥4. Myocardial ischemia was considered as mild when SSS was 4 to 8, moderate 9 to 13, and serious with SSS > 13 [27].

Vit D was measured by radioimmunoassay (RIA) (DIAsource 25OH Vitamin D total -RIA-CT Kit), which is an accurate and validated method. The specimen collection, preparation and procedure were performed according to the manufacturer instructions (DIAsource ImmunoAssays S.A). Patients with Vit D values < 10 ng/mL, 10–29 ng/mL and ≥30 ng/mL were considered to have a deficiency, insufficiency and sufficiency, respectively.

### 2.2. Statistical Analysis

MPI scores were evaluated with Vit D cutoff values of 10 ng/mL, and further with cutoff values of 20 ng/ml. Seasonal Vit D was separated into two groups. The first group was patients who had Vit D from September to February and the second group from March to August.

Nonparametric statistics were used because the Kolmogorov–Smirnov test revealed the deviation of the variables from the normal distribution. The Spearman rho correlation coefficient was used to assess association of Vit D levels with the SSS, SRS and SDS scores and age. The Mann-Whitney U-test was used to compare Vit D levels between the subjects with and without ischemia, and between men and women. Patients with normal MPI and no ischemia, were considered as control group. Backward stepwise multivariate logistic regression was used to identify the independent predictors of ischemia among Vit D, gender, and age. A statistical analysis was performed using SPSS software (version 23.0, IBM). Two-tailed values of *p* < 0.05 were regarded as statistically significant.

### 2.3. Ethics

This retrospective study was performed after approval by the Hospital’s Clinical Research Committee. Our Hospital’s Clinical Research Committee does not require written patient’s consent in retrospective studies. All collected data were coded with no patient personal information provided, and the statistical analysis was performed blinded.

## 3. Results

Among 113 MPIs, 58/113 (51.3%) patients had normal, and 55/113 (48.7%) patients had abnormal MPI. Figure 1 and Figure 2 show images of the myocardial perfusion schintigraphy of a normal and abnormal MPI, respectively. Their Vit D scores indicated Vit D deficiency in 20/113 (17.7%) patients, insufficiency in 86/113 (76.1%) and sufficiency in 7/113 (6.2%) patients. Amongst the 20 patients with Vit D deficiency, 16 demonstrated abnormal MPI (80%), and between the 86 patients with Vit D insufficiency, 38 exhibited abnormal MPI (44.2%), in contrast to only 1 in 7 patients with normal Vit D values that had an abnormal MPI (14.3%).

Correlation analysis showed a negative association of Vit D levels with SSS (rho = −0.232, *p* = 0.014) and SRS (rho = −0.250, *p* = 0.008). No association was found between the Vit D levels and SDS (rho = −0.003, *p* = 0.975) or age (rho = −0.140, *p* = 0.139). The Mann-Whitney U test showed that ischemia lowers the values of Vit D, (Table 1, Figure 3). There was no difference in Vit D levels between men and women. When applying a Vit D cutoff value of 20 ng/mL, there were 63/113 (55.75%) and 50/113 (44.25%) patients with less than and over 20 ng/mL Vit D, respectively. Among them, there were 39/63 (61.9%) patients with abnormal and 16/50 (32%) patients with normal MPI. Statistical analysis found no significance with any of the studied parameters. Interestingly, the analysis showed that Vit D and gender were independent risk factors for myocardial ischemia, with male gender at more risk compared to females.

## 4. Discussion

Cardiovascular disease consists of an important escalating challenge in European countries, responsible for approximately 45% of all deaths. Some countries, such as Greece, demonstrate a current failure to effectively adjust the modifiable risk factors leading to deaths secondary to stroke or ischemic heart disease. [33]

Although several noninvasive imaging tests exist to assess the status of myocardial function and of coronary arteries [34], MPI is widely available and reliable test with high sensitivity [35]. In most cases, an abnormal MPI indicates myocardial ischemia due to coronary artery disease. However, several other non-cardiac conditions may coexist, such as depression, anxiety, or Vit D deficiency, which may affect myocardial function and lead to an abnormal MPI without coronary artery disease [27,28].

In our study, most patients had either Vit D deficiency (17.7%) or insufficiency (76.1%), and very few (6.2%) had normal values of Vit D. In several studies, Vit D deficiency or insufficiency have been frequently reported in various countries. Thus, Vit D inadequacy has been reported in 36% of young adults and 57% of hospitalized patients in the United States [36]. In central Europe, Vit D levels have been reported on an average below the 30 ng/mL level [37]. In Greece, the Mediterranean paradox exists, consisting of a higher Vit D deficiency compared to other European countries, even though a high exposure to sunlight exists throughout the year. This paradox is unclear, but it has been attributed to various reasons including the low nutritional Vit D intake of the Greek population [38].

Vit D deficiency may increase the risk of osteoporosis, cancer [39] and cardiovascular conditions [40,41]. However, a systematic review and meta-analysis based on 51 clinical trials of moderate quality, failed to demonstrate a significant reduction in mortality and cardiovascular risk associated with Vit D [42]. In our study, we found a significant association of Vit D and myocardial ischemia, because low Vit D was an independent predictor of myocardial ischemia. Another recent study reported that low Vit D levels were associated with an increased risk of coronary artery disease [43]. Similarly, Vit D status in a cohort of 478 patients with acute myocardial infarction showed significantly low levels of Vit D in 431 of the 478 individuals [44]. Likewise, a study concluded that low serum Vit D was an independent risk factor for acute myocardial infarction, especially in patients with Vit D levels < 20 ng/mL [45]. In our study, the statistical analysis revealed no significant impact of a Vit D cutoff value of 20 ng/mL on myocardial ischemia.

In our study, no association was found between gender and Vit D levels, but male gender was an independent risk factor for myocardial ischemia. In contrast, a previous study reported that female gender was associated with increased severity of coronary artery disease and independently associated with significant Vit D deficiency [46]. However, several studies have reported that such variations between men and women in some studies may be due to variations in the presentation of the coronary artery disease in women, frequently with atypical symptoms, or with asymptomatic myocardial ischemia, leading to a delayed diagnosis of ischemia until when it is already advanced [47]. Additionally, compared with men, symptomatic women with angina and proven myocardial ischemia frequently have no significant coronary artery stenosis on angiography, but rather coronary microvascular dysfunction [48].,

In respect to seasonal variation, it was found in a group of patients with acute myocardial infarction that Vit D deficiency was less prominent during the period from July to the end of September, and it could be predicted by autumn/winter sampling [44]. In our cases, no statistical significance was found between the seasonal Vit D and myocardial ischemia.

Limitations of our study include its retrospective nature, with consequences consisted of incomplete medical history of participants and their physical characteristics. In addition, no information was available concerning the patients’ recent status, as well as further serum/plasma markers such as CK, CK-MB, cardiac troponin T, troponin I and myoglobin. In any event, our study demonstrated that Vit D levels had a negative association with SSS and SRS, indices associated with myocardial dysfunction. Since MPI SPECT represents a consistently reliable non-invasive imaging method, it may be used in individuals with low Vit D to rule out myocardial involvement. Further, prospective controlled studies are needed to verify our findings.

## 5. Conclusions

Low levels of vitamin D were associated with higher SSS and SRS, which indicate myocardial dysfunction. However, no statistical significance was noticed between the seasonal vitamin D levels and myocardial ischemia. Even though no association was found between gender and vitamin D levels, myocardial ischemia was more common in men compared to women. Further prospective, controlled studies are needed to verify our findings.

## Figures and Tables

**Figure 1 medicina-57-00774-f001:**
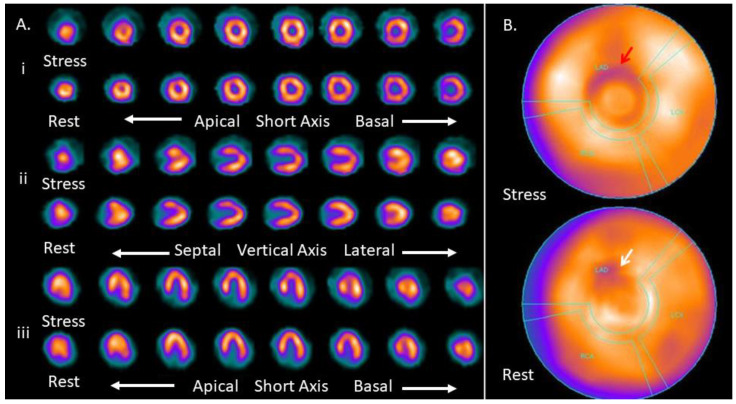
Normal MPI (a summed stress score less than 4). This image shows a normal MPI. The arrow points to a nonsignificant finding seen in the apical portion of the anterior wall of the left ventricle. The visual SSS was 1 (SSS < 4). (**A**) The three axis images show stress and rest MPI images in the short axis (i), vertical axis (ii) and horizontal axis (iii). (**B**) The bull’s eye image shows the entire myocardium. The red arrow shows the finding seen in the stress MPI, and the white arrow shows the improvement of the finding at rest.

**Figure 2 medicina-57-00774-f002:**
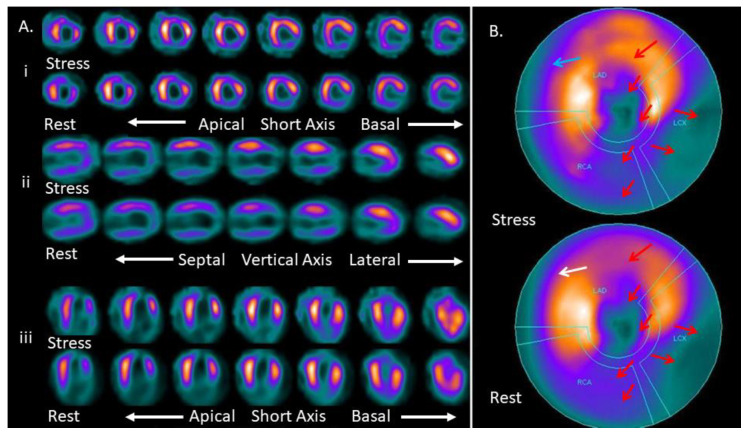
Abnormal MPI (non-reversible defects: a myocardial scar or severe ischemia). Hypoperfused areas are seen in the apex, inferior, inferior lateral, apical and middle part of the anterior wall (red arrows), and the basal part of the anterior-septal wall (blue arrow) of the left ventricle. (**A**) The three axis images show stress and rest MPI images in the short axis (i), vertical axis (ii) and horizontal axis (iii). The visual SSS was 33, the SRS was 32, and the SDS was 1, indicating a non-reversible myocardial ischemia. (**B**) The bull’s eye image shows the entire myocardium. The red arrow shows the hypoperfused areas seen in both the stress and rest MPI (non-reversible defects seen in myocardial infraction, or in severe ischemia); the blue arrow shows the defect in the anterior-septal wall, and the white arrow shows the reversibility in the rest MPI (SDS 1).

**Figure 3 medicina-57-00774-f003:**
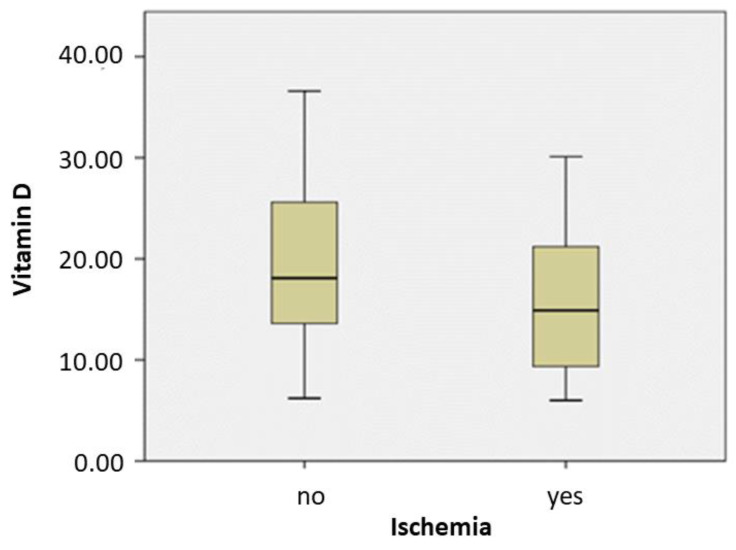
Box plots of the vitamin D distribution in subjects with and without ischemia. The logistic regression analysis showed that vitamin D (coefficient B = −0.068, *p*= 0.020, odds ratio = 0.934, confidence interval [0.882, 0.989]), and gender (coefficient B = −1.238, *p* = 0.003, odds ratio = 0.290, confidence interval [0.128, 0.654]) are independent predictors of ischemia.

**Table 1 medicina-57-00774-t001:** Descriptive statistics and differences of vitamin D levels between subjects with and without ischemia, and between men and women.

Variable	Groups	Median (Minimum, Maximum)	*p*-Value (Mann-Whitney U-Test)
ischemia	No	18.1 (6.2, 36.6)	0.013
	Yes	14.9 (6.0, 30.1)	
gender	Male	14.9 (6.0, 28.7)	0.153
	Female	17.4 (6.0, 36.6)	

## Data Availability

The data presented in this study are available on request from the corresponding author. The data are not publicly available due to privacy issues.

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
