# Peer review of "Vitamin D Deficiency as a Risk Factor for Myocardial Ischemia"

_medicina, 2021, doi:10.3390/medicina57080774_

Round 1

Reviewer 1 Report

In a retrospective study, Batsi and coauthors investigated if variations in the Vit D levels are linked to myocardial ischemia (MI).

The authors concluded that “there is a significant association of Vit D and MI, where low Vit D levels may represent a risk factor for MI. Furthermore, Vit D and gender were independently associated with MI .

This is an excellent study on an important topic. In general, the manuscript is well prepared and the study carefully designed, and easy to read. This study brings very important knowledge regarding the link between Vit D deficiency and MI. These results may trace pathways for better screening tools identifying patients at risk developing MI as a complement to traditional existing screening tools. While the findings are interesting and important to the field, I have some concerns detailed below:

General comment

Overall, the manuscript looks too short unless it is a Short communication paper. Did the authors conducted any follow up on patients at risk based on their findings?

Abstract

Line 19: please spell out MIP SPECT before using the acronym for the first time.

Line 30: remove extra space at the beginning of the sentence.

Introduction

The introduction could be improved from its current format with more background.

Line 43: please list the “other conditions”.

Materials and Methods

Patients demographics should be provided.

Line 62-65: this paragraph is a significant confounding factor that may significantly bias the results.

Line 68-70: what was to concordance between the two readers?

Line 80: …manufacturer instructions instead of manufacture instructions.

Line 88: add space between 20 and ng/ml

Line 87-90: Please moved to Materials and Methods section.

Line 99-103: numbering is wrong. Also, please moved the paragraph to Materials and Methods section.

Results

Line 105-109: This part should be moved to Materials and Methods section.

Data is convincing, but not well-presented. Can the authors break down data by gender, race and ethnicity? Can the authors present data in text in a graph format?

Could the authors confirm their findings by using traditional screening tools?

Discussion

Is the Greek population more at risk of MI than the rest of the European population?

Line 154: add space between < and 20 ng/ml.

Line 159-163: please fix formatting error.

Line 174: Remove extra space before “However”

Line 175: Remove extra space before “MPI SPECT”

References

Please fix formatting error for the following references:

18, line 230

19, line 232

29, line 261

30, line 264

31, line 267.

Figures

Please increase the font on Figures 1 and 2, it is hard to see.

Reviewer 2 Report

The title of the manuscript, “Vitamin D deficiency as a risk factor for myocardial ischemia”, was very impression for me in the beginning because I found a new marker to indicate myocardial ischemia. It’s pity that the experimental design as well as its results let me disappointed. As commonly known, the markers for cardiovascular diseases included CK, CK-MB, cardiac troponin T, troponin I, myoglobin, cardiac enzymes, but no data to show the positive control and analyze the potential relationship between or among the markers. My major question is 1) who is first? Cardiovascular diseases or VitD deficiency?  2) Do you think if you have to consider the relationship between VitD and other markers? In addition, I have some concerns below:

  1. Manuscript should intraduct the markers for cardiovascular diseases presented in serum/plasma such as included CK, CK-MB, cardiac troponin T, troponin I, myoglobin, cardiac enzymes.
  2. I checked the plagiarism and found it’s 40%, please make it decrease using your own words.
  3. Figure 1 & 2 should be quantitative and present the histogram.
  4. I disagree that the current data supported the correlation between VitD and cardiovascular diseases. Please provide more data or explain well about your current data.
  5. VitD is not specific marker for cardiovascular related diseases. Generally, Vitamin D deficiency can lead to a loss of bone density, which can contribute to osteoporosis and fractures (broken bones). It’s sure that severe vitamin D deficiency can also lead to other diseases. What is the authors’ consideration of the crosstalk among VitD, bone problems and cardiovascular disease?

Round 2

Reviewer 2 Report

The revised version is good to address my concerns. I do think the manuscript is ready to be publication and recommend editors accept the current version for publication!

Thanks!